# Improving Explanation Reliability through Group Attribution

## Abstract

Although input attribution methods are mainstream in understanding predictions of DNNs for straightforward interpretations, the non-linearity of DNNs often makes the attributed scores unreliable in explaining a given prediction, deteriorating the faithfulness of the explanation. However, the challenge could be mitigated by explaining groups of explanatory components rather than the individuals, as interaction among the components can be reduced through appropriate grouping. Nevertheless, a group attribution does not explain the component-wise contributions so that its component-interpreted attribution becomes less reliable than the original component attribution, indicating the trade-off of dual reliabilities. In this work, we first introduce the generalized definition of reliability loss and group attribution to formulate the optimization problem of the reliability trade-off. Then we specify our formalization to Shapley value attribution and propose the optimization method G-SHAP. Finally, we show the explanatory benefits of our method through experiments on image classification tasks.

## 1 Introduction

The advance in deep neural networks facilitates a training model to learn high-level semantic features in a variety of fields, but intrinsic difficulties in explaining predictions of DNNs become a primary barrier to real-world applications, especially for domains requiring trustful reasoning for model predictions.

While various approaches have been proposed to tackle the challenge, which includes deriving global behavior or knowledge of a trained model (Kim et al., 2018), explaining the semantics of a target neuron in a model, (Ghorbani et al., 2019; Simonyan et al., 2013; Szegedy et al., 2015), introducing self-interpretable models (Zhang et al., 2018; Dosovitskiy et al., 2020; Touvron et al., 2020; Arik & Pfister, 2019), input-attribution methods became the mainstream of post-hoc explanation methods since they explain a model prediction by assigning a scalar score to each explanatory component (feature) of its input data, yielding the straightforward explanation for end-users through data-corresponded visualization such as a heatmap.

However, since each explanatory component is explained with a single scalar score, the nonlinearity in DNNs makes their scores less reliable in explaining a model's prediction. It results in the discrepancy between the explained and actual model behavior for a prediction, deteriorating the faithfulness of the explanation.

As it is the inherent challenge of input attribution methods, the problem has been studied and tackled with various approaches and perspectives: (Grabisch & Roubens, 1999) formalizes the axiomatic interactions for cooperative games, (Tsang et al., 2018) explains the statistical interaction between input features from learned weights in DNN, (Kumar et al., 2021) introduces Shapley Residuals to quantify the unexplained contribution of Shapley values, (Janizek et al., 2021) extends Integrated Gradients (Sundararajan et al., 2017) to Integrated Hessians to explain the interaction between input features.

While these approaches have improved the explainability to the DNN's nonlinearity, their explaining scores are not corresponded to each explanatory components in many cases, reducing the interpretability of explanations.

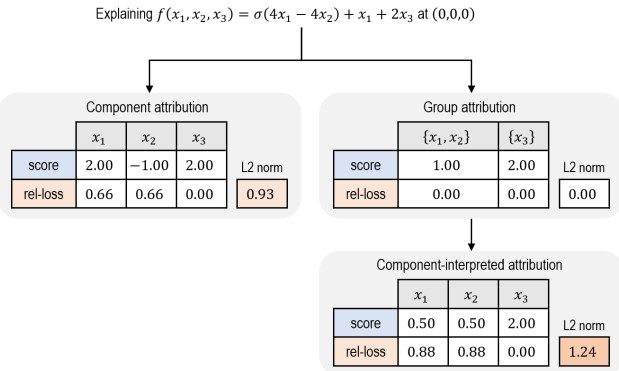

Figure 1: Trade-off of the dual reliability loss of a group attribution for a simple non-linear function. Grouping $x_1, x_2$ resolves their interaction so that it reduces the reliability loss of the group $\{x_1, x_2\}$ but increases those of component-interpreted scores. Here attribution score and its reliability loss are defined as the input gradient and expected L2 error of its tangent approximation, which are $\frac{\partial}{\partial x_i} f(\mathbf{x})$ and $\mathbb{E}_{t \sim \mathcal{N}(0,1)}[(f(\mathbf{x} + t\mathbf{e}_i) - f(\mathbf{x}) - t\phi_i)^2]$, respectively.

Instead, it can be alleviated by explaining a model's prediction in terms of groups explanatory components rather than the individuals, termed **group attribution**. Appropriate grouping can weaken the interaction among the components, yielding more reliable explanation.

However, a group attribution does not attribute scores to the individual components so that interpreting a group attribution in terms of the individual components results in less reliable explanation than the original component attribution.

Therefore, both group-wise and component-interpreted attribution reliability should be considered for deriving a group attribution, implying a trade-off optimization problem. Figure 1 illustrates this problem with simple a non-linear function.

In this paper, we present our work as follows: In Section 2, we introduce the generalized definition of reliability loss and group attribution to formulate the optimization problem of the reliability trade-off. In section 3, we integrate our formalization with Shapley value attribution (Lundberg & Lee, 2017) and propose the grouping algorithm G-SHAP. We choose the Shapley value as our scoring policy for two reasons: 1) it has been utilized as a popular attribution method for its model-agnostic characteristic and well-founded axiomatic properties. 2) it becomes less reliable when there are strong interactions among the explanatory component's contribution, as it take the aggregation of the contributions over all coalition states. In section 4, we show the explanatory benefits of our method through experiments on image classification tasks as follows: 1) we verify the grouping effect of G-SHAP through quantitative and visual analysis. 2) we validate our grouping approach by comparing it with several baseline grouping method which would yield the similar grouping result to ours. 3) we show the improvement in local explainability of a prediction through the estimation game, which utilizes the deletion game (Petsiuk et al., 2018; Wagner et al., 2019) to measure the error of model output changes.

Our contributions are summarized as follows:

1. We introduce two novel concepts to improve the limited reliability of input attribution methods: **reliability loss** that quantifies the discrepancy between the explained and the actual model behavior for a prediction, **group attribution** that explains a prediction in terms of groups of explanatory components. Since a group attribution becomes less reliable in explaining component-wise contributions, we formulate the optimization problem to resolve the reliability trade-off. While we choose the Shapley value as our scoring policy, our formulation consists of generalized terms, applicable for other input attribution methods.

2. We propose **G-SHAP**, a grouping algorithm for Shapley value attribution. We empirically show that G-SHAP has better local explainability of a model prediction than SHAP. We also validate the effectiveness of our grouping approach by comparing it with several baseline grouping methods, which would yield the similar grouping results to ours.

## 2 GENERAL FORMALIZATION FOR RELIABILITY LOSS AND GROUP ATTRIBUTION

### 2.1 RELIABILITY LOSS OF AN ATTRIBUTION

Let $y^* = f(\mathbf{x}^*)$ be a model prediction to explain, where $\mathbf{x}^* = (x_1^*, ..., x_N^*) \in \mathbb{R}^N$ and $f : \mathbb{R}^N \to \mathbb{R}$ are the input data and the model function, respectively. Let $\Phi$ be attributing (scoring) function that takes a model function $f$ and a target input $\mathbf{x}^*$ and returns the attribution scores $\boldsymbol{\phi} \in \mathbb{R}^N$. As we consider $f$ and $\mathbf{x}^*$ is fixed, we introduce the translated model function $f^*(\mathbf{x}) = f(\mathbf{x} + \mathbf{x}^*)$ to simplify our henceforth definitions. Since explaining $f(\mathbf{x})$ at $\mathbf{x} = \mathbf{x}^*$ is equivalent to explaining $f * (\mathbf{x})$ at $\mathbf{x} = \mathbf{0}$, we have

$$\boldsymbol{\phi} = (\phi_1, ..., \phi_N) = \Phi(f, \mathbf{x}^*) = \Phi(f^*, \mathbf{0}) \in \mathbb{R}^N \tag{1}$$

Similarly, let $\Xi$ be a function that quantifies the reliability loss in explaining the prediction with an arbitrary attribution $\mathbf{a} = (a_1, ..., a_N) \in \mathbb{R}^N$ as below.

$$\xi(\mathbf{a}) = \Xi(f^*, \mathbf{a}) \geq 0 \tag{2}$$

where the lower value implies the more reliable attribution in explaining the prediction. We note that $\mathbf{a} = (a_1, ..., a_N)$ can be arbitrary, not necessarily $\boldsymbol{\phi}$.

### 2.2 GROUP ATTRIBUTION AND ITS RELIABILITY LOSS

A group attribution attributes a score to each group of explanatory components, where the components within each group are treated as one shared variable. Formally, let $\mathbf{G} = \{G_1, ..., G_M\}$ be a grouping (partition) of the component set $X = \{x_1, ..., x_N\}$. Then the group-mapped function $f_{\mathbf{G}}^*$ assigns each group-variable $g_i$ to its corresponding components variables of $X$, defined as

$$f_{\mathbf{G}}^*(g_1, ..., g_M) = f^*(g_{\sigma(1)}, ..., g_{\sigma(N)}) \tag{3}$$

where $\sigma$ is the group map such that $x_i \in G_{\sigma(i)}$ for each $1 \leq i \leq N$. For example, the group-mapped function of of $f(a, b, c)$ with the grouping $\mathbf{G} = \{\{a, b\}, \{c\}\}$ is $f_{\mathbf{G}}(g_1, g_2) = f(g_1, g_1, g_2)$.

Once we have a grouping $\mathbf{G}$, its group attribution $\boldsymbol{\phi}_{\mathbf{G}}$ is defined as the attribution scores of $f_{\mathbf{G}}^*$, which is

$$\boldsymbol{\phi}_{\mathbf{G}} = (\phi_{G_1}, ..., \phi_{G_M}) = \Phi(f_{\mathbf{G}}^*, \mathbf{0}) \in \mathbb{R}^M \tag{4}$$

By definition, each group score $\phi_{G_j}$ indicates the co-contribution of their components $x_i \in G_j$. Note that it is not necessarily equal to the sum of the component scores in general.

Similarly, we can derive the reliability loss of a group attribution $\boldsymbol{\phi}_{\mathbf{G}}$ as below,

$$\xi(\boldsymbol{\phi}_{\mathbf{G}}) = \Xi(f_{\mathbf{G}}^*, \boldsymbol{\phi}_{\mathbf{G}}) \tag{5}$$

which tells that how reliable in explaining the prediction with the group attribution. From the definition, we can say a group attribution $\boldsymbol{\phi}_{\mathbf{G}}$ is more reliable than the component attribution $\boldsymbol{\phi}$ if $\xi(\boldsymbol{\phi}_{\mathbf{G}}) < \xi(\boldsymbol{\phi})$ and less reliable if $\xi(\boldsymbol{\phi}_{\mathbf{G}}) > \xi(\boldsymbol{\phi})$.

### 2.3 COMPONENT-INTERPRETATION OF A GROUP ATTRIBUTION AND ITS RELIABILITY LOSS

As discussed in the introduction, a group attribution does not attribute scores to their belonging components so that its component-interpreted scores would have larger reliability loss than the original component attribution. To address this, we need to first formalize the **score-interpreting function** $\zeta$, which interprets a group attribution $\boldsymbol{\phi}_{\mathbf{G}}$ in terms of the individual components, denoted with the tilde as below.

$$\widetilde{\boldsymbol{\phi}}_{\mathbf{G}} = (\widetilde{\phi}_1, ..., \widetilde{\phi}_N) = \zeta(\boldsymbol{\phi}_{\mathbf{G}}) \in \mathbb{R}^N \tag{6}$$

It is notable that the interpreting policy can vary depending on the component or score's semantics but must not utilize any information of the prediction. For example, defining $\zeta(\boldsymbol{\phi}_{\mathbf{G}}) = \boldsymbol{\phi}$ is not acceptable.

Consequently, the component-wise reliability loss of a group attribution $\boldsymbol{\phi}_{\mathbf{G}}$ is given as below.

$$\xi(\widetilde{\boldsymbol{\phi}}_{\mathbf{G}}) = \Xi(f^*, \widetilde{\boldsymbol{\phi}}_{\mathbf{G}}) \tag{7}$$

### 2.4 FORMULATING THE OPTIMIZATION PROBLEM OF THE RELIABILITY TRADE-OFF

In order to formalize the trade-off of dual reliability of a group attribution, we first normalize the improvement and deterioration of the reliability losses.

As the reliability loss of a group attribution $\xi(\phi_{\mathbf{G}})$ is expected to be lower than the that of the original component attribution $\xi(\phi)$, we consider $\xi(\phi)$ as the baseline and define the normalized score for a group attribution's reliability **(NGR)** $\mathcal{G}$ as the ratio of the improved amount to the baseline, given as

$$\mathcal{G}(\mathbf{G}) = \frac{\xi(\phi) - \xi(\phi_{\mathbf{G}})}{\xi(\phi)} \tag{8}$$

It follows that higher $\mathcal{G}$ is better: it becomes 1 if $\xi(\phi_{\mathbf{G}}) = 0$ (maximum improvement) and 0 if $\xi(\phi_{\mathbf{G}}) = \xi(\phi)$ (no improvement). It can be negative if the grouping is ill-chosen.

On the other hand, the component-interpreted reliability loss of a group attribution $\xi(\widetilde{\phi}_{\mathbf{G}})$ is expected to be higher than the original $\xi(\phi)$. Since the most uninformative grouping $\mathbf{G}_{\text{all}}$ that merges all components into one group i.e., $\mathbf{G}_{\text{all}} = \{G_1\} = \{\{z_1, ..., z_N\}\}$ is expected to have the largest component-interpreted reliability loss, we define the normalized score for a group attribution's component-interpreted reliability **(NCR)** $\mathcal{C}$ as the ratio of the less-deteriorated (saved) amount to $\mathbf{G}_{\text{all}}$ to the gap, given as

$$\mathcal{C}(\mathbf{G}) = \frac{\xi(\widetilde{\phi}_{\mathbf{G}_{\text{all}}}) - \xi(\widetilde{\phi}_{\mathbf{G}})}{\xi(\widetilde{\phi}_{\mathbf{G}_{\text{all}}}) - \xi(\phi)} \tag{9}$$

It also follows that higher $\mathcal{C}$ is better: it becomes 1 if $\xi(\widetilde{\phi}_{\mathbf{G}}) = \xi(\phi)$ (no deterioration), 0 if $\xi(\widetilde{\phi}_{\mathbf{G}}) = \xi(\widetilde{\phi}_{\mathbf{G}_{\text{all}}})$ (deteriorated as $\mathbf{G}_{\text{all}}$).

Since there are two singular cases that should be avoided for searching the grouping, which are singleton grouping (no-grouping) and the all-grouping $\mathbf{G}_{\text{all}}$. As their $(\mathcal{G}, \mathcal{C})$ scores are $(0, 1)$ and $(1, 0)$, respectively, we define the optimization objective $\mathcal{L}$ as the geometric mean of two scores, given as

$$\mathcal{L}(\mathbf{G}) = \max\left\{\frac{\mathcal{G}(\mathbf{G}) + \epsilon}{1 + \epsilon}, 0\right\}^{\frac{1}{2} - \beta} \max\left\{\frac{\mathcal{C}(\mathbf{G}) + \epsilon}{1 + \epsilon}, 0\right\}^{\frac{1}{2} + \beta} \tag{10}$$

where $\epsilon \geq 0$ is the tolerance hyperparameter for dealing with negative $\mathcal{G}$ and $\mathcal{C}$ values and $\beta \in [-1/2, 1/2]$ is the balancing hyperparameter that positive $\beta$ weighs more to $\mathcal{C}$ than $\mathcal{G}$ and vice versa. It follows that larger $\mathcal{L}$ implies better group attribution.

## 3 APPLICATION TO SHAPLEY VALUE

### 3.1 SHAPLEY VALUE AND ITS RELIABILITY LOSS

Shapley value has originated from cooperative game theory, indicating the fair division of given reward to each player. It has been utilized as the axiomatic attribution method for post-hoc model explanations (Lundberg & Lee, 2017), where the players and the reward are corresponded to the binary explanatory components and the output difference of the model prediction, respectively.

Formally, let $Z = \{z_1, ..., z_N\}$ be the set of binary explanatory components and $\mathcal{Z} = \{0, 1\}^N$ be the set of the all possible coalition states $\mathbf{z} = (z_1, ..., z_N)$, where each $z_i = 1, 0$ indicates whether $z_i$ is involved in the coalition or not, respectively. Once a model function $f : \mathcal{Z} \to \mathbb{R}$ is given, **contribution** of $z_i$ at a coalition state $\mathbf{z} \in \mathcal{Z}$ is defined as

$$h_i(\mathbf{z}) = f(\mathbf{z}_{i=1}) - f(\mathbf{z}) \tag{11}$$

where the notation $\mathbf{z}_{i=1}$ means $\mathbf{z}$ is assigned with $z_i = 1$. Since $h_i(\mathbf{z})$ is trivially zero when $z_i = 1$, we restrict the domain of $h_i$ to $\mathcal{Z}_{i=0} := \{\mathbf{z} \in \mathcal{Z} | z_i = 0\}$.

Then **Shapley value** of $z_i$ is given as the weighted sum of contributions $h_i$ at all possible coalition states, which is

$$\phi_i = \sum_{\mathbf{z} \in \mathcal{Z}_{i=0}} w_N(|\mathbf{z}|) h_i(\mathbf{z}), \quad w_N(k) = \frac{k!(N - k - 1)!}{N!} \tag{12}$$

where $|\mathbf{z}|$ is termed coalition size, the number of 1s in $\mathbf{z}$.

Since it follows that $\sum_{\mathbf{z} \in \mathcal{Z}_{i=0}} w_N(|\mathbf{z}|) = 1$, a Shapley value $\phi_i$ can be considered as the expected value of $h_i$ by regarding the weights $w_N(|\mathbf{z}|)$ as the probability, i.e., $\phi_i = \mathbb{E}_{\mathcal{Z}_{i=0}}[h_i]$. This perspective of defining Shapley values naturally leads to measure the expected L2 error of the contributions, given as

$$\xi_i^2(a_i) = \mathbb{E}_{\mathcal{Z}_{i=0}}[(h_i - a_i)^2] = \sum_{\mathbf{z} \in \mathcal{Z}_{i=0}} w_N(|\mathbf{z}|)(h_i(\mathbf{z}) - a_i)^2 \tag{13}$$

which is named **Shapley error** of attributing $z_i$ with a score $a_i$. Now we define the reliability loss of Shapley attribution as

$$\xi^2(\mathbf{a}) = \sum_{i=1}^N \xi_i^2(a_i) \tag{14}$$

where $\boldsymbol{\phi} = (\phi_1, ..., \phi_N)$ and $\mathbf{a} = (a_1, ..., a_N)$. Similar to the property of ordinary mean and variance, it follows that $\xi_i^2(a_i) = \xi_i^2(\phi_i) + (a_i - \phi_i)^2$ so that the reliability loss consequently satisfies

$$\xi^2(\mathbf{a}) = \xi^2(\boldsymbol{\phi}) + \|\mathbf{a} - \boldsymbol{\phi}\|_2^2 \tag{15}$$

which implies that the reliability loss is minimized to $\xi^2(\boldsymbol{\phi})$ when $\mathbf{a} = \boldsymbol{\phi}$, showing the optimality of Shapley value attribution.

## 3.2 SHAPLEY GROUP ATTRIBUTION

Let $\mathbf{G} = \{G_1, G_2, ..., G_M\}$ be a partition (grouping) of the component set $Z = \{z_1, ..., z_N\}$ with non-empty groups. Then a group-wise coalition state $\mathbf{z} \in \mathcal{Z}$ under the grouping $\mathbf{G}$ is restricted to the cases that components in each group $G_j$ are all involved or not, denoted as $\mathbf{z}[G_j] = 1, \mathbf{z}[G_j] = 0$, respectively. Since each group $G_j$ has a binary involvement state, the coalition state has $M$ degree of freedom and can be represented as a $M$-dimensional binary vector.

First, contribution of $G_j$ at a coalition state $\mathbf{z} \in \mathcal{Z}$ is defined as the output difference of $f$ by switching all $z_i \in G_j$ to 1, given as

$$h_{G_j}(\mathbf{z}) = f(\mathbf{z}_{G_j=1}) - f(\mathbf{z}) \tag{16}$$

where the notation $\mathbf{z}_{G_j=1}$ means that $\mathbf{z}$ is assigned with $z_i = 1$ for all $z_i \in G_j$. Similar to $h_i$, we consider the domain of $h_{G_j}$ as $\mathcal{Z}_{G_j=0}$, defined as $\mathbf{z} \in \mathcal{Z}$ satisfying $\mathbf{z}[G_j] = 0$ and $\mathbf{z}[G_m] \in \{0, 1\}$ for all $1 \leq m \neq j \leq M$.

Consequently, the Shapley value and error of a group $G_j$ is defined as

$$\phi_{G_j} = \mathbb{E}_{\mathcal{Z}_{G_j=0}}[h_{G_j}], \qquad \xi_{G_j}^2(a_{G_j}) = \mathbb{E}_{\mathcal{Z}_{G_j=0}}[(h_{G_j} - a_{G_j})^2] \tag{17}$$

It is notable that the expectation operation $\mathbb{E}_{\mathcal{Z}_{G_j=0}}$ is not compatible with the component-wise case $\mathbb{E}_{\mathcal{Z}_{i=0}}$ since the dimension of coalition states chagned from $N$ to $M$, which implies that Shapley value of a group does not equal to the sum of its components' Shapley values.

## 3.3 G-SHAP: ALGORITHM FOR SHAPLEY GROUP ATTRIBUTION

Since the number of groups $M$ can vary from $[1, N]$, it is difficult to evaluate the group-wise Shapley terms from component-wise terms. Moreover, group-wise coalition states depend on not only the number of groups but also the grouping itself. It implies that the optimization problem is more challenging than the set-partitioning problem, where the target value of each subset is fixed.

However, as (Guanchu, 2022) has shown the effectiveness, Shapley statistics can be approximated by excluding the components or groups which have little effect on the target. Despite the incompatibility of Shapley weights, the weight decomposition property $w_{N-1}(k) = w_N(k) + w_N(k+1)$

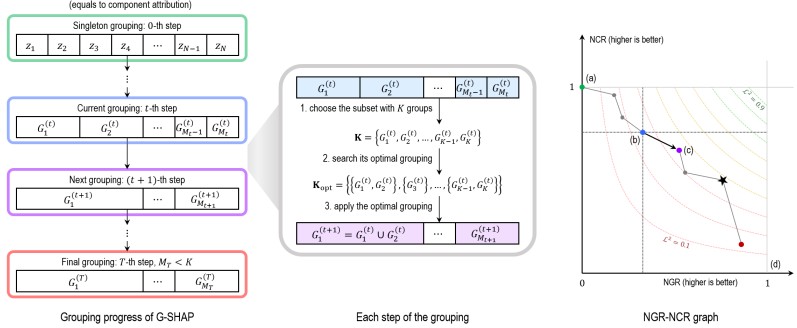

Figure 2: Illustration of the G-SHAP algorithm with corresponding NGR-NCR graph: (a) initial stage: grouping starts with the singleton grouping (equal to component-attribution), where $(\mathcal{G}, \mathcal{C}) = (0, 1)$, (b) an arbitrary middle step, (c) the very next step: the optimal grouping of the core subset $K$ is applied (d) last step: remaining number of group is less than the core set size. Dotted curve indicates contour line of the objective $\mathcal{L}$ and star mark indicates the best Shapley group attribution which G-SHAP finally returns.

suggests that $\phi_i$ and $\xi_i^2$ can be decomposed with $z_j$-conditioned contributions as

$$
\begin{aligned}
\phi_i &= \sum_{l=0}^{N-2} \sum_{|\mathbf{z}|=l} w_N(l) h_i(\mathbf{z}_{j=0}) + w_N(l+1) h_i(\mathbf{z}_{j=1}) \\
\xi_i^2 &= \sum_{l=0}^{N-2} \sum_{|\mathbf{z}|=l} w_N(l)(h_i(\mathbf{z}_{j=0}) - \phi_i)^2 + w_N(l+1)(h_i(\mathbf{z}_{j=1}) - \phi_i)^2
\end{aligned}
\tag{18}
$$

where $z_j \neq z_i$ can be chosen arbitrary. It implies that if $h_i(\mathbf{z}_{j=0}), h_i(\mathbf{z}_{j=1}) \approx h_i(\mathbf{z})$ then $\phi_{i|j=0}, \phi_{i|j=1} \approx \phi_i$ and $\xi_{i|j=0}, \xi_{i|j=1} \approx \xi_i$ so that the appropriate exclusion would yield more accurate approximation for Shapley statistics.

In our method G-SHAP, we take $\epsilon_i = \sum_{j \neq i}(\phi_{i|j=0} - \phi_i)^2 + (\phi_{i|j=1} - \phi_i)^2$ the heuristic for the exclusion. Components or groups with top-$k$ $\epsilon_i$ values are considered as the **core subset K** of given grouping **G**. Once we have **K** then we observe all binary states $\{0, 1\}^k$ of **K** to derive the optimal grouping of **K**, where the excluded components or groups are fixed. Then we apply the optimal grouping and continue the progress until $|\mathbf{G}| < k$. Overall progress is illustrated in the Figure 2.

## 4    EXPERIMENTAL RESULTS

While there have been existing group or cluster-wise explanation methods (Masoomi et al., 2020; Singh et al., 2018) their grouping criteria and objective are different from ours so that comparing those explanations with ours would not be resonable. Therefore, as mentioned in the introduction, we have focused on verifying the explanatory benefits of our group attribution (G-SHAP) through comparison with the corresponding component attribution (SHAP).

### 4.1    EXPERIMENTAL SETUP

As mentioned in the introduction, we have applied the proposed method on the validation datasets of Flower5(multi-class) (Mamaev, 2018), MS COCO 2014(multi-label) (Lin et al., 2014), and Pascal VOC 2012(multi-label) (Everingham et al., 2012) with ImageNet 2012Russakovsky et al. (2015) pretrained ResNet-50 (He et al., 2016) model, where Flower5 stands for subset of Flower dataset with 5 distinctive classes (daisy, dandelion, rose, sunflower, tulip). We have fine-tuned the model to each datasets as the average prediction accuracy is 94.7%, 82.9%, and 90.3%, respectively. For MS COCO and Pascal VOC models, we have taken logit value of top-1 label as the output.

Since our heuristic of choosing the core subset requires conditional Shapley values which have $\mathcal{O}(N^2)$ terms, we have considered superpixels of a image as explanatory components. We have experimented with two superpixel methods, quick-shift (Vedaldi & Soatto, 2008) and graph-based

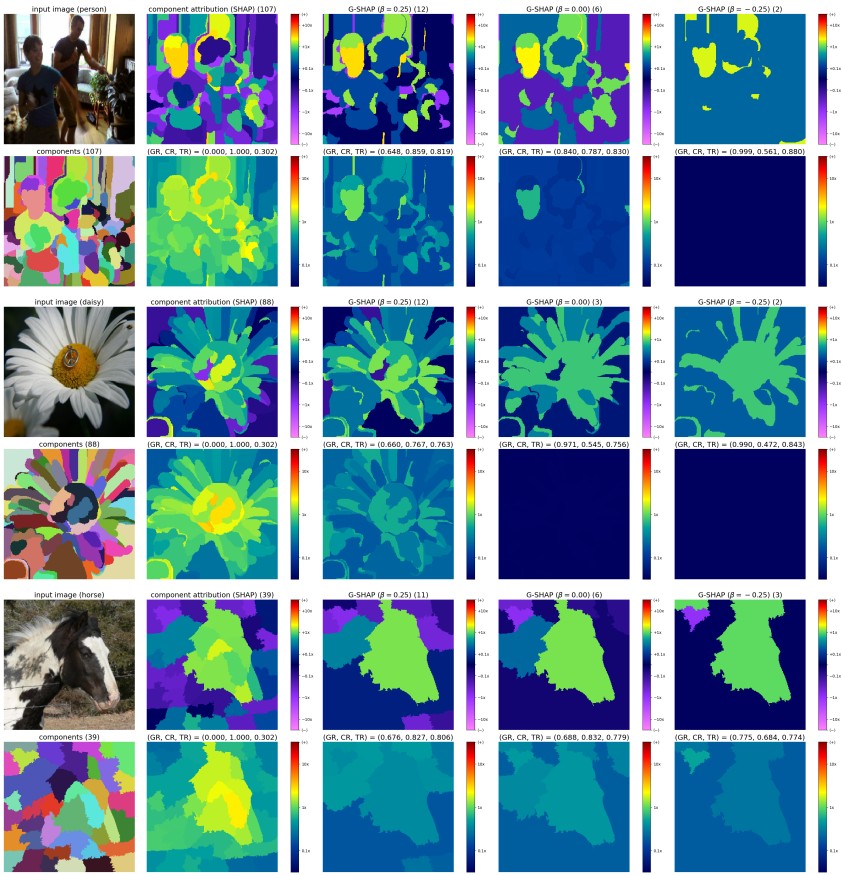

Figure 3: G-SHAP results for the image classification task, taken from MS COCO, Flower5, Pascal VOC dataset, where superpixels are chosen as graph-based, graph-based, and quick-shift, for each image, respectively. The 3-5th columns stand for $\beta = 0.25, 0.00$ and $-0.25$, respectively. For each image, heatmap of the upper row indicates the attribution score and the lower row indicates the attribution reliability. Heatmaps are area-normalized ratio to their base values, which are their sum divided by entire area of the image.

(Felzenszwalb & Huttenlocher, 2004) segmentation method, which existing attribution methods LIME (Ribeiro et al., 2016) and XRAI (Kapishnikov et al., 2019) use. As Shapley value considers binary input states, we have defined the input map as mean-color masking function so that $z_i = 0, 1$ corresponds to the mean-colored and the original superpixel, respectively. We also define the score estimating function $\zeta$ as distributing a group score according to pixel area, i.e., $[\xi(\phi_{\mathbf{G}})]_j = \frac{w_j}{\sum_{z_j \in G} w_j} \phi_G$ for $z_j \in G$. We set core subset dimension $k = 10$, $\beta = 0.0$, and $\epsilon = 0.1$ as default.

## 4.2 OPTIMIZATION EFFECTS OF G-SHAP

We have observed NGR, and NCR to verify the improved and saved Shapley error of G-SHAP attribution for $\beta = -0.25, 0.00, 0.25$, stated at Table 1 (left). As NGR, NCR indicates normalized ratio of the amounts, it tells that $75\% \sim 91\%$ of the baseline $\Xi(\phi)$ are resolved through grouping while $62\% \sim 68\%$ of the bound gap $\Xi(\widetilde{\phi}_{\mathbf{G}_{\text{all}}}) - \Xi(\phi)$ is saved for $\beta = 0.00$ case. It has also been observed $\beta$ considerably affect the NGR and NCR scores such that positive $\beta$ weighs NCR much than NGR, whereas the negative $\beta$ weighs NCR much than NGR, agreeing with our expectation.

Figure 3 shows that the balancing effect can also be verified qualitatively, telling that higher $\beta$ results in higher NCR that the heatmap of G-SHAP is closer to the component attribution (SHAP) but lower NGR that Shapley errors are less improved. On the other hand, lower $\beta$ results in the opposite as well

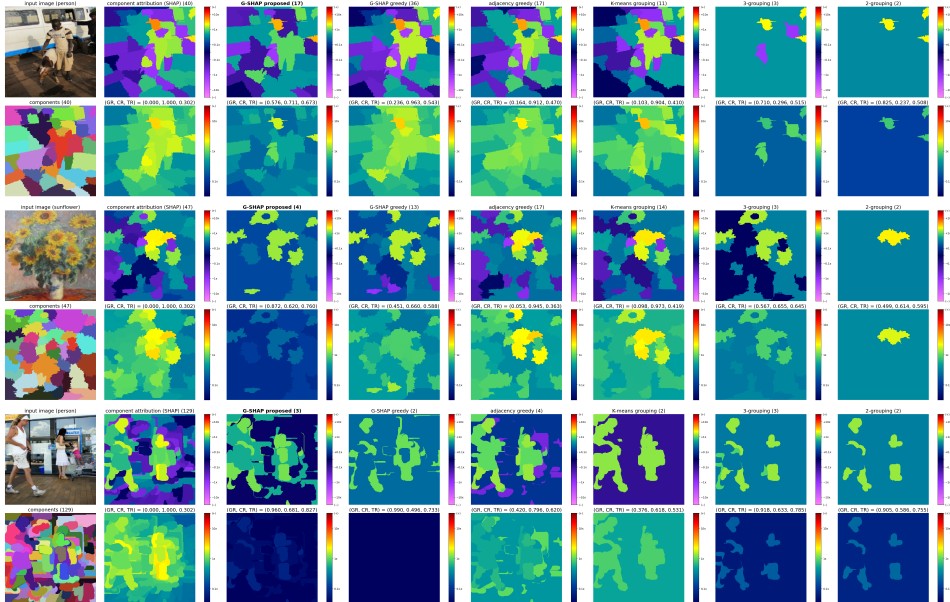

Figure 4: Comparison results of G-SHAP with various heuristic methods, where the images are taken from COCO, Flower5, and Pascal VOC dataset, and the superpixels are chosen from quick-shift, quick-shift, and graph-based method, respectively.

| Superpixels | Datasets | $\beta = 0.25$ | | $\beta = 0.00$ | | $\beta = -0.25$ | | Methods | Quick-Shift | | Graph-based | |
|---|---|---|---|---|---|---|---|---|---|---|---|---|
| | | NGR | NCR | NGR | NCR | NGR | NCR | | NGR | NCR | NGR | NCR |
| Quick-shift | COCO | 0.596 | 0.832 | 0.799 | 0.679 | 0.939 | 0.667 | 2-grouping | 0.786 | 0.552 | 0.818 | 0.558 |
| | Flower5 | 0.593 | 0.831 | 0.758 | 0.642 | 0.919 | 0.616 | 3-grouping | 0.696 | 0.646 | 0.754 | 0.635 |
| | VOC | 0.602 | 0.829 | 0.824 | 0.670 | 0.942 | 0.680 | K-means grouping | 0.476 | 0.701 | 0.493 | 0.646 |
| Graph-based | COCO | 0.704 | 0.779 | 0.892 | 0.630 | 0.973 | 0.692 | Adjacency greedy | 0.443 | 0.720 | 0.423 | 0.711 |
| | Flower5 | 0.603 | 0.779 | 0.840 | 0.605 | 0.950 | 0.653 | G-SHAP greedy | 0.678 | 0.584 | 0.648 | 0.561 |
| | VOC | 0.719 | 0.776 | 0.905 | 0.626 | 0.975 | 0.667 | G-SHAP porposed | 0.794 | 0.664 | 0.879 | 0.620 |

Table 1: Reliability scores of the G-SHAP for $\beta = 0.25, 0.00$ and $-0.25$ (left) and comparison with baseline heuristics, where scores are averaged on the datasets (right)

but also tells that G-SHAP attribution consists of a few groups with salient superpixels. It implies that our method needs to be compare with baseline grouping methods to validate our approach as the sanity-check, discussed in the later subsection.

### 4.3 VALIDATING THE GROUPING APPROACH

In order to show the validity of our grouping strategy, we have compared G-SHAP with various grouping heuristics which would likely yield the similar results, stated in the Table 1 (right) and illustrated in the Figure 4. The details for each grouping heuristics are described below.

First, we have employed **2-grouping** and **3-grouping** methods as G-SHAP with lower $\beta$ yields few groups. The 2-grouping method sorts the components by Shapley values and splits them into two groups by merging the top $1 \le k \le N$ components and the others, and returns the best one. Similarly, the 3-grouping method considers all grouping cases with of top-$k$, bottom-$m$, and the middle-$(N - k - m)$ components and picks the best one. While the results show that their NGR and NCR are slightly lower than ours (less than $0.1$), their explanation contains too minimal information since the most intermediate-salient superpixels are neglected.

We have also employed **K-means grouping** and **Adjacency-greedy grouping** methods to test the grouping performance of merging components with closer attribution scores, as the salient superpixels of G-SHAP attribution are often attributed with higher or closer Shapley values. The K-means grouping method clusters of the components into $2 \le k \le 10$ groups and returns the best grouping, where the distance metric is given as the difference of normalized Shapley value (divided by superpixel area). The **Adjacency-greedy** method iteratively merges two groups with the closest

| Superixel method | Attribution | min-deletion | | | max-deletion | | | random-deletion | | |
|---|---|---|---|---|---|---|---|---|---|---|
| | | Flower5 | COCO | VOC | Flower5 | COCO | VOC | Flower5 | COCO | VOC |
| Quick-shift | SHAP | 0.476 | 0.566 | 0.671 | 0.444 | 0.438 | 0.515 | 0.446 | 0.465 | 0.587 |
| | G-SHAP | 0.171 | 0.158 | 0.204 | 0.159 | 0.144 | 0.191 | 0.147 | 0.131 | 0.169 |
| Graph-based | SHAP | 0.532 | 0.674 | 0.786 | 0.458 | 0.494 | 0.563 | 0.469 | 0.537 | 0.646 |
| | G-SHAP | 0.223 | 0.176 | 0.223 | 0.191 | 0.161 | 0.190 | 0.177 | 0.149 | 0.180 |

Table 2: Estimation game result of component attribution (SHAP) and group attribution (G-SHAP)

normalized Shapley values, and returns the best grouping. As these strategies are expected to retain higher NCR scores, both methods show that NCR is simlar or slightly higher than ours, whereas NGR is clearly lower than ours. It implies that these two heuristics could not resolve group-wise Shapley errors as any interaction statistics are utilized.

In addition, we have employed G-SHAP without core subset searching as the ablation study, named **G-SHAP greedy**, which instead greedily merges two groups which are expected to improve the $\mathcal{L}$ the most. As it shows that both NGR and NCR are around 0.1 lower than our method in average, implying that the optimization problem is challenging to solve with simple greedy approach so that optimal partition searching in the core subset is necessary.

## 4.4 ESTIMATION GAMES

Deletion game (Petsiuk et al., 2018; Wagner et al., 2019) is the main strategy to assess the attribution scores, which removes each component of input data in sequence and evaluates the model output drop through AUC of the curve. However, these methods usually rely on the ranking of the attribution scores so that it does not assess the reliability of the attribution in general. Therefore, we have employed this idea in a different way, termed **estimation game**, which aims to measure the error of expected output changes to the actual one. As Shapley value indicates the expected model contribution, this assessment approach is intuitive to understand has also been utilized in (Guanchu, 2022). For the deletion process, we have employed three types of deletions: min-deletion, max-deletion, and random-deletion, which deletes (fills mean-color) inputs in increasing, decreasing, and random order of attribution score, respectively. Since model logits can be arbitrarily scaled depending on the prediction, we have normalized as follows: (1) we have linearly rescaled $y$-axis such that $y = 0, 1$ stands for ground image (mean-colored image) and target image, respectively. (2) we have also linearly rescaled $x$-axis as it indicates the ratio of removed pixels to entire pixels. Therefore removal game always starts from $(0, 1)$ and

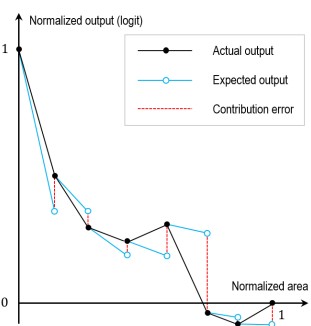

Figure 5: Illustration of the estimation game, measuring the error of expected drop to the actual drop

ends with $(1, 0)$, illustrated in the Figure 5. As shown in Table 2, G-SHAP resolves around 60% to 70% of the L2 estimation error of component attribution (SHAP), providing a better understanding of the local behavior of the model.

## 5 CONCLUSION

Though input-attribution methods provide clear interpretation as the explanations correspond to the data, non-linearity of deep models intrinsically hinders reliability of attribution. In this work, we have presented novel perspective of quantifying the reliability, attributing groups, and formulating it with the optimization problem. We have chosen Shapley value for the scoring policy to specify the terms and propose the grouping algorithm G-SHAP. We have shown the explanatory benefits of our group attribution in multiple perspectives. Its improvement of a group attribution's reliability loss is clearly larger than deterioration of component-interpreted reliability loss, and also improved local explainability of a model's prediction. However, since our method utilizes Shapley conditional terms and search partition spaces with iteration, its computation cost is too high to start with pixel-wise components. Deeper analytical approach and utilizing the prior information of input components would improve the performance and feasibility, left as potential for future works.

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
