# OpenReview forum: "Improving Explanation Reliability through Group Attribution"
_ICLR.cc/2023/Conference — Submitted to ICLR 2023_

### Official Review · Reviewer_ssJm · 2022-10-19

**Confidence:** 4
**Correctness:** 3
**Technical Novelty And Significance:** 3
**Empirical Novelty And Significance:** 1
**Recommendation:** 6

**Clarity, Quality, Novelty And Reproducibility:**

**Clarity**

The writing of the paper can be improved. In particular, the introduction section and the formulation of reliability and group attribution are confusing and need more clarity.

**Novelty**

The paper introduces formulations for reliability loss and group attributions.

**Reproducibility**

The code for G-SHAP is shared in an anonymous repo for reproducibility.

**Strength And Weaknesses:**

**Strengths**

1. A new perspective to attributing the importance of input features using *reliability loss* that indicates the discrepancy between predicted and actual contributions and *group attribution* that attributes scores to a group of explanations instead of the individuals.

2. The proposed G-SHAP outperforms vanilla SHAP feature attribution methods and other baseline attribution methods across three real-world datasets.

**Weaknesses and Open Questions**

1. The paper argues about the usability of the explanations in Section 1 but does not discuss it further in the empirical analysis. Further, is usability directly related to the empirical performance of an explanation?
2. The definitions of reliability loss and group attribution are agnostic to the choice of explanation method, but the paper limits its application to just Shapeley values.
3. What is the intuitive interpretation of a group? What does it represent? Is it just a simple grouping of individual attributions, or does it aim to group components with similar attributions?
4. The paper does not compare the results with state-of-the-art feature attribution methods like GradCAM, Integrated Gradients, LIME, etc.
5. One main takeaway of the paper is that a model prediction is better understood with groups than individual components. The authors motivate the problem using the non-linearity of the model. In addition, grouping multiple features should intuitively lead to better performance as they may break the correlation of dependent features. It would be great if the authors could comment on this.

**Summary Of The Paper:**

With the increasing use of machine learning models, it becomes important to understand their decisions before using them in high-stakes real-world applications. Despite the effectiveness of feature-attribution methods in explaining model decisions, the non-linearity of ML models hampers the reliability of the explanations. In this work, the authors hypothesize that the reliability of feature attribution scores can be improved by attributing scores to groups instead of individual features. In particular, they present a novel perspective of group attribution, propose an optimization problem, and integrate it with Shapley value attribution to propose G-SHAP. Qualitative and quantitative empirical results on image classification tasks show that model predictions can be better understood with groups than with individual components.

**Summary Of The Review:**

The paper presents a new perspective to attribute group features but lacks clarity and empirical evidence using state-of-the-art methods.

---

> ### Author Response · Authors · 2022-11-19
> **Authors' response for Reviewer ssJm**
>
> ```
> The paper argues about the usability of the explanations in Section 1 but does not discuss it further in the empirical analysis. Further, is usability directly related to the empirical performance of an explanation?
> ```
>
> We apologize for the unclarity. The "usability" means "interpretability", the ease of understanding the attributed explanation. It is the major strength of input attribution methods as the end-user can easily understand data-corresponded scores through heatmap visualization. We intended to tell that the mentioned explanation methods can better explain a model's prediction through providing more information such as interactions but retrieve explanation scores that would not directly correspond to each explanatory component, reducing the interpretability. Therefore, the term does not indicate the evaluation metric of an explanation. We refined the corresponding part in the updated paper to make it clear.
>
> ```
> The definitions of reliability loss and group attribution are agnostic to the choice of explanation method, but the paper limits its application to just Shapley values.
> ```
>
> We choose the Shapley value as our scoring policy for two reasons: (1) it has been utilized as a popular attribution method for its model-agnostic characteristic and well-founded axiomatic properties. (2) it becomes less reliable when there are strong interactions among the explanatory component's contribution, as it take the aggregation of the contributions over all coalition states.
> However, it does not mean that only Shapley value is applicable in our formulation. As it is inherent challenge for input attribution methods to explain each explanatory component's non-linear model influence with a scalar score, we consider applying the group attribution to the other input attribution methods.
>
> ```
> What is the intuitive interpretation of a group? What does it represent? Is it just a simple grouping of individual attributions, or does it aim to group components with similar attributions?
> ```
>
> A group attribution is defined as the component attribution of its group-mapped function, which treats components in each group as one shared variable (Eq. 3). Therefore, a group score can intuitively be interpreted as the co-influence of the belonging components to the model predicton, under the same scoring policy.
>
> For the Shapley value case, Shapley value of a group can be interpreted as the expected value (weighted sum) of the group-wise contribution (output difference caused by participating all players in the group). Moreover, a group score is not necessarily equal to the sum of its components' scores.
>
> ```
> The paper does not compare the results with state-of-the-art feature attribution methods like GradCAM, Integrated Gradients, LIME, etc.
> ```
>
> Since our work aims to improve the explanation reliability of attribution methods through grouping explanatory components, we focused on showing the explanatory benefits of our group attributions in comparison with the corresponding component attribution (SHAP).
>
> ```
> One main takeaway of the paper is that a model prediction is better understood with groups than individual components. The authors motivate the problem using the non-linearity of the model. In addition, grouping multiple features should intuitively lead to better performance as they may break the correlation of dependent features. It would be great if the authors could comment on this.
> ```
>
> We sincerely thank for the refinement. As the statement is clear and well explains our research goal, we applied it in the updated paper.

---

### Official Review · Reviewer_FuXt · 2022-10-19

**Confidence:** 3
**Correctness:** 3
**Technical Novelty And Significance:** 3
**Empirical Novelty And Significance:** 2
**Recommendation:** 3

**Clarity, Quality, Novelty And Reproducibility:**

- Originality: The work definitely seems original to the best of my knowledge.
- Quality and Clarity are significant areas of improvement for this work.
- Reproducibility: The authors provide code which seems considerably detailed, with notes and notebooks. I haven't had the chance to run the code myself.

**Strength And Weaknesses:**

Strengths:
- The paper provides and motivates an important problem. i.e. group attributions are important to capture "groupings"/non-linear interactions of features as they contribute to a prediction, something that component-wise attribution methods lack. On the other hand, just providing group attributions can result in in obfuscating contributions of individual components. Hence the trade-off that forms the basis of this paper.

Weaknesses:
1. The paper is extremely hard to read. Is littered with typos, unfinished sentences and overloaded notation. On average, there's a typo in every other sentence, too many for me to list here. The experiment section specially has too many typos to even count. I strongly advise the authors to read through there paper multiple times and improve its readability and remove typos. Even if the paper has an extraordinary contribution it becomes really hard to feel excited about it if it looks like it has been put together last minute.

some examples:
-- right above eq (2) "and an given scores" (remove "an")
-- above eq (7) "an score estimation" ("an" --> "a" is redundant)
-- above eq (11) "with their Once we have the" (sentence doesn't make sense)
-- section 3 1st line "to each player involved in." (unfinished sentence)

2. There are decisions being made here that are not clearly motivated, there's an overall lack of theoretical justification for decisions, though heuristic justifications can be made. For example:
- under eq(3) the logic behind the conclusion that $a = \xi$ is the global minimum of eq(3) doesn't make sense. Was this intended to be $a = \phi$? . Why were equations (2) and (3) formulated the way they are formulated? It'd help if the line of thinking behind these equations is clearly articulated before the equations are presented.
- Before equation (10), a seemingly arbitrary decision is made about the upperbound for CR when, as the authors say it doesn't have one. What's the reason behind choosing this particular upperbound?
- below equation (8), \psi can be arbitrarily defined, how arbitrary? does a \psi that simply spits out random numbers work? e.g. it's unclear what the equivalent \psi is in the G-SHAP formulation.

3. The experiment section, in addition to be seemingly very rushed, also seems undercooked in terms of comparisons. There are no comparisons with other group-wise feature selection approaches (e.g. see "instance-wise feature grouping" by Masoomi et al. NeurIPS 2020 and the methods they compare against) and even the results compared to baselines don't show much improvement.

**Summary Of The Paper:**

The paper proposes a generalizable optimization strategy that codifies a way to trade-off between group attribution and component-wise attributions calculated from explanation methods. The end product is a group attribution version of a given explainer (they propose G-SHAP as an example for groupwise SHAP) that balances the reliability loss from a group attribution compared to component attribution. They also provide formalization for reliability loss (both component-wise and group-wise) with the total loss defined as the geometric mean of the two.

**Summary Of The Review:**

This is a promising line of work overall. But in my opinion is not quite ready for publication. The authors should consider taking their time and improving the motivation, writing and experiments provided in the paper. The paper in its current form will be a disservice to their own work if it gets published as it is.

---

> ### Author Response · Authors · 2022-11-19
> **Authors' response for Reviewer FuXt**
>
> We appreciate your insightful comments and feedback. We answered each of the addressed concerns below.
>
> ```
> Under eq(3) the logic behind the conclusion that is the global minimum of eq(3) doesn't make sense. Was this intended to be $\phibf$? Why were equations (2) and (3) formulated the way they are formulated? It'd help if the line of thinking behind these equations is clearly articulated before the equations are presented.
> ```
>
> We apologize for the typo, and it should be $\mathbf{a} = \mathbf{\phi}$. We intended to state that any component-interpreted group attribution would be less reliable than the original component attribution. We deleted the statement in the updated paper because scoring function $\mathrm{\Phi}$ and the loss measure $\mathrm{\Xi}$ must be specified to prove it, which should be discussed in Section 3.
>
> For equation (2), (3), we intended to introduce a reliability loss function which evaluates each score-wise reliability loss and aggregates to the attribution-wise reliability (total reliability loss in the draft paper). We deleted the L2 norm definition of the attribution-wise reliability as it considers the general case of input attribution methods.
>
> ```
> Before equation (10), a seemingly arbitrary decision is made about the upperbound for NCR when, as the authors say it doesn't have one. What's the reason behind choosing this particular upperbound?
> ```
>
> The component-interpreted reliability is expected to increase if components or groups are merged into fewer groups because the merged groups contain less information about their individual component scores than before. It implies that the component-interpreted reliability loss of the all-merged grouping is expected to be the upper bound among all grouping cases. In our case, it is indeed the true upper bound. As the reason is not well-described in the draft paper, we revised the corresponding part in the updated paper.
>
> ```
> below equation (8), $\zeta$ can be arbitrarily defined, how arbitrary? does a $\zeta$ that simply spits out random numbers work? e.g. it's unclear what the equivalent $\zeta$ in the G-SHAP formulation.
> ```
>
> $\zeta$ is the function that provides the component-wise interpretation for a group-wise attribution, whose definition depends on the component or score's semantics. Purposeless $\zeta$ would provide meaningless component-wise interpretation as well, yielding unintended group attribution. In our work, we defined $\zeta$ as dividing each group-wise Shapley value $\phi_G$ to their belonging components $z_j \in G$ with weights $w_j$, i.e., $\frac{w_j}{\sum_{z_j \in G}{w_j}}\phi_G$, where the weight is defined as the superpixel area. We clarified the definition of $\zeta$ in the updated paper.
>
>
> ```
> The experiment section, in addition to be seemingly very rushed, also seems undercooked in terms of comparisons. There are no comparisons with other group-wise feature selection approaches (e.g. see "instance-wise feature grouping" by Masoomi et al. NeurIPS 2020 and the methods they compare against) and even the results compared to baselines don't show much improvement.
> ```
>
> We sincerely thank for addressing the reference. We did not include other grouping or clustering-based explanation methods for the comparison because their grouping criteria and objective are different from ours so that it is difficult to compare the grouping quality of those methods and ours with the same metric. Instead, we focused on showing the explanatory benefit of our group attributions in comparison with the corresponding component attribution (SHAP). We first verified the grouping effect of our method through NGR, NCR scores and visual analysis, showing that our group attribution resolved about 80\% of the reliability loss of SHAP while deteriorated about 35\% of the reliability loss compared to the all-grouped case (9\% deterioration if compared to the SHAP's loss).
> Second, we validated our grouping approach by showing the limited grouping performance of several baseline methods, which would likely yield a similar grouping to ours. We also compared our method with the greedy version of G-SHAP to verify the effectiveness of the proposed core subset searching strategy.
> Finally, we showed that our method improves the local explainability of a prediction through the estimation game, which measures not AUC but error of model output in the sequential deletion of input features. Since Shapley value indicates the fairly-divided contribution to the model output, lower error implies more reliable Shapley-valued attribution. Similar idea was utilized in Yang Liu et al., (NeurIPS'21) and Guanchu et al., (ICML'22).

---

> > ### Comment · Reviewer_FuXt · 2022-11-19
> > **Response to Authors**
> >
> > Thank you for responding. I will take this response into consideration when providing final recommendation.

---

### Official Review · Reviewer_boPc · 2022-10-25

**Confidence:** 3
**Correctness:** 4
**Technical Novelty And Significance:** 3
**Empirical Novelty And Significance:** 3
**Recommendation:** 6

**Clarity, Quality, Novelty And Reproducibility:**

Novel. However parts of the submission are less clear due to significant amount of mathematical notation.

**Strength And Weaknesses:**

+ves:

- A really good idea to automate the computation of group attributions of feature subsets that interact. Also using the G-SHAP heuristic to determine the optimal subset combination is novel.

-ves:

- Empirical results are shown for images alone, it would be good to show these on tabular / text datasets too.
- The disentangling of group to individual features is less clear from the text and experiments in the sense as to how these differ from actual shapley values of all individual features and how the repeat problem of assigning individual attribution values within each subset does not occur.
- Its unclear if the shapley axioms hold at group level based on the computed group attributions.
- It might be good to build experiments where feature interactions/dependence are known a priori and evaluate if the computed subsets match ground truth.

**Summary Of The Paper:**

This work proposes a new explanation algorithm called G-SHAP that computes attributions of feature subsets and attributions of individual features within each subset. The motivation to do so is to overcome the problem of interactions between features, thereby grouping features that interact together into the same group.

**Summary Of The Review:**

See above.

---

> ### Author Response · Authors · 2022-11-19
> **Authors' response for Reviewer boPc**
>
> We appreciate your insightful comments and feedback. We answered each of the addressed concerns below.
>
> ```
> Empirical results are shown for images alone, it would be good to show these on tabular / text datasets too.
> ```
>
> We agree with the concern and consider more experiments in the suggested domains.
>
> ```
> The disentangling of group to individual features is less clear from the text and experiments in the sense as to how these differ from actual shapley values of all individual features and how the repeat problem of assigning individual attribution values within each subset does not occur.
> ```
>
> We apologize for the unclarity. In our work, the score interpreting function $\zeta$ divides each group-wise Shapley value $\phi_G$ to their belonging components $z_j \in G$ with weights $w_j$, i.e., $\frac{w_j}{\sum_{z_j \in G}{w_j}}\phi_G$, where the weight is defined as the superpixel area.
> For the second question, let $\mathbf{G}=\{G_1,...,G_M\}$ be a grouping and $\mathbf{K} \subseteq \mathbf{G}$ be the core subset to analyze. In order to search the optimal grouping of $\mathbf{K}$, we observe all possible binary states $\{0,1\}^{|\mathbf{K}|}$ of the core subset with fixing the other $M - |\mathbf{K}|$ states. As this progress is not well-described in the draft paper, we revised the corresponding section in the updated paper.
>
> ```
> It's unclear if the Shapley axioms hold at group level based on the computed group attributions.
> ```
>
> As we introduced in Section 2.2, a group attribution is defined as component attribution of its group-mapped function, which treats components in each group as one shared variable (Eq. 3). Since only the target function is replaced, it still follows the axioms of attribution methods.
>
> ```
> It might be good to build experiments where feature interactions/dependence are known a priori and evaluate if the computed subsets match ground truth.
> ```
>
> We agree with the suggestion and consider experiments on synthetic datasets, where groundtruth group structure is known.

---

### Official Review · Reviewer_BDjb · 2022-10-26

**Confidence:** 3
**Correctness:** 3
**Technical Novelty And Significance:** 3
**Empirical Novelty And Significance:** 3
**Recommendation:** 5

**Clarity, Quality, Novelty And Reproducibility:**


Clarity
The presentation of the manuscript is relatively clear and its content has a good flow.

Quality
The proposed method is sound and well motivated.
The evaluation could be improved.


Novelty
The proposed combination and reliability terms is, to the best of my knowledge, novel.

Reproducibility
There seem to be plans to release the code used for the experiments reported in the manuscript. This is a good step towards enabling reproducibility.

**Strength And Weaknesses:**

Strengths
- The proposed ideas seem novel to me
- Varied set of datasets
- Code to be released
- An ablation study is present.

Weaknesses
- High computational costs
- Focus on a single task
- Evaluation is not complete.

**Summary Of The Paper:**

The manuscript proposes to consider a reliability loss term to consider the differences between component-level explanations, group-level explanations and the actual explanations.

Based on this idea, an optimization approach ( G-SHAP) tailored for Shapley values is proposed.

Experiment on the image classification task on 4 datasets show the effectiveness of the proposed method

**Summary Of The Review:**


The proposed combination and reliability terms is sound and well motivated. To the best of my knowledge this is novel.
The manuscript has a good balance of verbal and formal descriptions.
The evaluation covers a good amount of different datasets which could help paint a wider picture of the capabilities of the proposed method.

My main concerns with the manuscript are the following:

As admitted in the paper, the proposed method seems to be computationally expensive, reason for which it might not scale to dense inputs, e.g. pixel-level attribution.

In Sec. 2.3, it is stated "it is required not to utilize any information of given prediction, or the estimated score would contain unexpected information of the actual component-wise contribution"
Could you further elaborate on this statement?

Regarding the grouping policies, is there a principled manner to select the grouping policy?

Currently the proposed method is only tested on the image classification task. It would strengthen the manuscript if pointers on how the proposed method could be applied to other tasks were provided.

Independently of its inner-workings, at the end of the day the proposed method is an explanation method. In this regard, it would have strengthen the manuscript if the proposed method was tested under the sanity checks proposed by [Adebayo et al., NeurIPS'18]. This way there could be guarantees on whether the generated heatmaps/attribution maps do constitute valid explanations.

Very related to the previous point, the manuscript effectively show how integrating reliability and group attribution helps getting better attribution maps. However, it is hard to assess at this points where the proposed method is positioned, w.r.t. to the large pool of existing explanation methods.
Here I wonder if applying the suggested optimization to a state of the art explanation method would further improve its performance, or whether is this just the case for Shapley-based methods..

---

> ### Author Response · Authors · 2022-11-19
> **Authors' response for Reviewer BDjb**
>
> We appreciate your insightful comments and feedback. We answered each of the addressed concerns below.
>
> ```
> As admitted in the paper, the proposed method seems to be computationally expensive, reason for which it might not scale to dense inputs, e.g. pixel-level attribution.
> ```
>
> In our work, we did not consider prior knowledge of input components for the G-SHAP algorithm. We consider for the general case of Shapley value attribution so that we evaluated conditional Shapley values to choose the more adaptive core subset for searching, which improved the grouping performance but increased the computational cost.
> However, as dense inpust such as pixels or timesteps generally have the locality and other prior knowledge, we consider to utilize such information to reduce the computational cost of our method.
>
> ```
> In Sec. 2.3, it is stated "it is required not to utilize any information of given prediction, or the estimated score would contain unexpected information of the actual component-wise contribution" Could you further elaborate on this statement?
> ```
>
> We apologize for the unclarity. $\zeta$ is the score-interpreting function that provides the component-wise interpretation for a group-wise attribution. Its definition can vary depending on the component or score's semantics but should not utilize any information of the original component-wise attribution. For example, if one defines $\zeta$ as retrieving the original component-wise attribution regardless of the grouping, i.e., $\zeta(\mathbf{a}_{\mathbf{G}})=\mathbf{a}$, then the NCR score would always be 1. We revised the statement in the updated paper.
>
> ```
> Regarding the grouping policies, is there a principled manner to select the grouping policy?
> ```
>
> As the semantics and properties of attributed scores vary depending on the attribution methods, it does not have a principled way. Moreover, a grouping algorithm can be improved by utilizing the specific properties of given attribution method. For example, if merging components or groups does not affect the others, one can take parallel or divide-and-conqueror approaches for grouping.
>
> ```
> Currently the proposed method is only tested on the image classification task. It would strengthen the manuscript if pointers on how the proposed method could be applied to other tasks were provided.
> ```
>
> We agree with the concern and consider more experiments in other tasks, where obtaining reliable attribution is crucial.
>
> ```
> Independently of its inner-workings, at the end of the day the proposed method is an explanation method. In this regard, it would have strengthen the manuscript if the proposed method was tested under the sanity checks proposed by [Adebayo et al., NeurIPS'18]. This way there could be guarantees on whether the generated heatmaps/attribution maps do constitute valid explanations.
> ```
>
> We sincerely thank for the suggestion. We consider applying the suggested sanity-check method to validate of our methods.
>
> ```
> Very related to the previous point, the manuscript effectively show how integrating reliability and group attribution helps getting better attribution maps. However, it is hard to assess at this points where the proposed method is positioned, w.r.t. to the large pool of existing explanation methods. Here I wonder if applying the suggested optimization to a state of the art explanation method would further improve its performance, or whether is this just the case for Shapley-based methods.
> ```
>
> As it is inherent challenge for input attribution methods to explain each explanatory component's non-linear model influence with a scalar score, applying our formulation to other input attribution methods could improve the reliability of attributed explanations.

---

> > ### Comment · Reviewer_BDjb · 2022-11-29
> > **thanks for the feedback**
> >
> > I thank the authors for the provided feedback.
> > I will consider the provided clarifications in my final recommendation.

---

### Author Response · Authors · 2022-11-19
**Thank you for the constructive reviews**

We sincerely thank all reviewers for their valuable and insightful feedback. We revised the paper to improve the addressed concerns, including the following list of major changes.

1. We revised the manuscript for better readability, correcting typos and grammatical errors.
2. We refined the notations and figures to understand the proposed concepts better, including the followings.
* We disuse the score-wise reliability loss and their L2 aggregation in Section 2 because their definition relies on the scoring policy.
* We changed terminology "estimated" to "interpreted", applied to "component-estimated reliabilty loss" to "component-interpreted reliability loss" and "estimating function" to "interpreting function" since using both expression might confuse the readers.
* We added NGR-NCR graph to help understand G-SHAP progress for each step.
* We modified Section 3 in order to provide clear understanding for Shapley group terms and G-SHAP progress.
* We added the Figure 5, which intuitively illustrates the estimation game.
3. We clarified the reason why we chose Shapley values as the scoring policy in the introduction.
4. We clarified the reason why we compare G-SHAP and SHAP instead of existing attribution methods in the introduction.
5. We revised the experimental section to reflect our intentions and contributions more clearly.

We hope our revision and responses address all the reviewers’ concerns.
Best regards, Authors.

---

### Decision · Program_Chairs · 2023-01-20

**Decision:**

Reject

**Justification For Why Not Higher Score:**

Paper has promising ideas but the empirical study is under developed and needs to be significantly expanded to be convincing.

**Justification For Why Not Lower Score:**

N/A

**Metareview: Summary, Strengths And Weaknesses:**

This work proposes G-SHAP, a new explanation algorithm that computes attributions of feature subsets and attributions of individual features within each subset. This is motivated by the interactions between features--- it aims to improve the explanation reliability by grouping features that interact together into the same group.
Strength:
The paper studies an important problem.
The idea of computing attribution to feature groups and using reliability loss to optimize the grouping of features is interesting and novel

Weakness:
The paper felt under developed with several limitations in its current form.
Proposed approach is computationally expensive. Authors hinted at leveraging spatial structure in images to reduce computation but it is only a vague idea, not implemented.
Evaluation is incomplete. The  method is applied only to the image domain despite the method being promoted as a general approach. Some of the design choices seem to be image specific - like the score estimation function introduced in section 4.1. it also missed some relevant baselines.